# Antibiotic Overprescribing among Neonates and Children Hospitalized with COVID-19 in Pakistan and the Implications

**DOI:** 10.3390/antibiotics12040646

**Published:** 2023-03-24

**Authors:** Zia UI Mustafa, Amer Hayat Khan, Sabariah Noor Harun, Muhammad Salman, Brian Godman

**Affiliations:** 1Discipline of Clinical Pharmacy, School of Pharmaceutical Sciences, Universiti Sains Malaysia, Gelugor 11800, Penang, Malaysia; 2Department of Pharmacy Services, District Headquarter (DHQ) Hospital, Pakpattan 57400, Pakistan; 3Institute of Pharmacy, Faculty of Pharmaceutical and Allied Health Sciences, Lahore College for Women University, Lahore 54000, Pakistan; 4Strathclyde Institute of Pharmacy and Biomedical Science (SIPBS), University of Strathclyde, Glasgow G4 0RE, UK; 5Department of Public Health Pharmacy and Management, School of Pharmacy, Sefako Makgatho Health Sciences University, Pretoria 0208, South Africa; 6Centre of Medical and Bio-Allied Health Sciences Research, Ajman University, Ajman P.O. Box 346, United Arab Emirates

**Keywords:** neonates, child, COVID-19, hospitals, anti-infective agents, AWaRe classification, bacterial infections, antimicrobial resistance, Pakistan

## Abstract

There are concerns with excessive antibiotic prescribing among patients admitted to hospital with COVID-19, increasing antimicrobial resistance (AMR). Most studies have been conducted in adults with limited data on neonates and children, including in Pakistan. A retrospective study was conducted among four referral/tertiary care hospitals, including the clinical manifestations, laboratory findings, the prevalence of bacterial co-infections or secondary bacterial infections and antibiotics prescribed among neonates and children hospitalized due to COVID-19. Among 1237 neonates and children, 511 were admitted to the COVID-19 wards and 433 were finally included in the study. The majority of admitted children were COVID-19-positive (85.9%) with severe COVID-19 (38.2%), and 37.4% were admitted to the ICU. The prevalence of bacterial co-infections or secondary bacterial infections was 3.7%; however, 85.5% were prescribed antibiotics during their hospital stay (average 1.70 ± 0.98 antibiotics per patient). Further, 54.3% were prescribed two antibiotics via the parenteral route (75.5%) for ≤5 days (57.5), with most being ‘Watch’ antibiotics (80.4%). Increased antibiotic prescribing was reported among patients requiring mechanical ventilation and high WBCs, CRP, D-dimer and ferritin levels (*p* < 0.001). Increased COVID-19 severity, length of stay and hospital setting were significantly associated with antibiotic prescribing (*p* < 0.001). Excessive antibiotic prescribing among hospitalized neonates and children, despite very low bacterial co-infections or secondary bacterial infections, requires urgent attention to reduce AMR.

## 1. Introduction

In Pakistan, cases of COVID-19 have now been reported in almost every geographical location of the country and across all age groups. In Lahore, 19,367 children between the ages of 1 and 18 had been affected with COVID-19 between March 2020 and March 2021 [1], and in Islamabad, 5792 children up to 10 years of age had tested positive for COVID-19 from the start of the pandemic until April 2021 [2].

An early study from Karachi indicated that among children admitted with COVID-19, 26% of the cases were moderate and 12% severe, with a mortality rate of 7%, principally among those with severe disease [3]. A World Health Organization (WHO)-sponsored study also showed high mortality rates (up to 14%) among children in Pakistan admitted with moderate-to-severe COVID-19 during the early stages of the pandemic [4]. 

Whilst children typically have lower infection rates with COVID-19 than adults, alongside milder symptoms [5,6,7,8], up to 10% of children admitted to hospital with COVID-19 experienced severe disease. Up to 20% or more of admitted children were subsequently admitted to pediatric intensive care units (PICUs) [5,9,10,11]. Viral pneumonia was the commonest presenting condition [10,12,13]. However, there are concerns with children developing Kawasaki Disease (KD)-like symptoms, now known as Pediatric Inflammatory Multisystem Syndrome (PIMS-TS), alongside experiencing coagulation disorders as well as respiratory changes and abdominal involvement with COVID-19 [14,15,16,17,18,19,20]. These combined symptoms increase admission to (PICUs). Admission to PICUs is an issue in lower- and middle-income countries (LMICs) versus higher-income countries, with more limited healthcare resources and deaths among these hospitalized children [11]. This was reflected by the high mortality rates (40%) among pediatric patients admitted to Dr. Cipto Mangunkusumo Hospital in Indonesia at the start of the COVID-19 pandemic, before improvements in knowledge and possible treatments [21]. Admission to ICUs is a concern among patients with COVID-19, as they can develop fungal or bacterial co-infections, increasing mortality and hospital length of stay [22,23]. 

Whilst local health authorities in Pakistan and others have supported the prescribing of antibiotics in patients with COVID-19 with secondary bacterial or bacterial co-infections [10,24,25,26,27,28], their excessive use is common among patients across countries hospitalized with COVID-19, including children, especially with concerns with pneumonia and the implications [9,10,29,30,31]. This is despite only a limited number of patients actually having secondary bacterial or bacterial co-infections in practice [29,31,32]. These prescribing habits have not been helped by some national guidelines advocating the empiric prescribing of antibiotics along with hydroxychloroquine in patients with COVID-19, despite limited evidence of benefit [12,33,34,35]. Published studies have also documented appreciable prescribing of antibiotics in patients with COVID-19 in Pakistan, again, despite limited evidence of bacterial co-infections and secondary bacterial infections [23,36,37,38,39]. 

Excessive prescribing of antibiotics is a concern, as their inappropriate and unjustified use increases antimicrobial resistance (AMR) [40,41,42,43,44,45]. Increasing AMR appreciably increases morbidity, mortality and costs, and has been described as the next pandemic unless addressed [42,46,47,48,49]. This is a key issue in patients with COVID-19, as AMR rates are increasing with an appreciable number of inappropriate prescriptions [50,51,52]. Consequently, these inappropriate prescribing habits need to be curtailed where possible. This is particularly important in Pakistan, which is currently the third largest consumer of antibiotics worldwide, with an appreciable increase in antibiotic use in recent years [53]. Alongside this, both multi-drug resistance (MDR) and extensive drug resistance (XDR) bacteria have been seen in different regions of Pakistan [54,55,56,57,58,59].

A number of published studies from Pakistan have highlighted appreciable prescribing of antibiotics in adult hospitalized patients with COVID-19. This is despite the limited prevalence of bacterial co-infections or secondary bacterial infections [37,38,39]. In addition, there has been considerable prescribing of ‘Watch’ and ‘Reserve’ antibiotics from the WHO AWaRe list. This is important as the antibiotics from the ‘Watch’ and ‘Reserve’ list have a greater resistance potential than those from the ‘Access’ list [41,60]. Their prescribing in Pakistan needs to be appreciably reduced to achieve a suggested target of 60% ‘access’ antibiotics being prescribed [37,38,39,60]. However, we are unaware of any studies that have been conducted among neonates and children in Pakistan to investigate the current utilization patterns in this group. This is important given concerns with mortality rates among children with COVID-19 admitted initially to hospitals in Pakistan, as well concerns generally with children admitted to PICUs in LMICs with COVID-19 [3,11].

Consequently, the aims of this study were to thoroughly investigate the clinical manifestations and laboratory findings of neonates and children admitted to hospitals in Pakistan with COVID-19. Secondly, to investigate the prevalence of bacterial co-infections and secondary bacterial infections, alongside the extent of antibiotic prescribing among this population hospitalized in Pakistan due to COVID-19. Subsequently, to use the findings to guide future strategies in Pakistan if there is excessive prescribing of antibiotics, especially ‘Watch’ and ‘Reserve’ antibiotics. 

## 2. Results

A total of 1237 neonates and children were seen in the emergency rooms of participating hospitals between March 2022 and December 2022. Among these children, 511 were admitted to the COVID-19 wards. Seventy-eight cases were subsequently excluded due to incomplete medical records. This left a final sample of 433 children (Figure 1). 

The demographic details of the children are shown in Table 1. Most of the study population were children aged 5 to 12 years (45%), followed by those aged 1 to 5 years (33.7%), and 64% were male children. Of the admitted children, 85.9% were COVID-19-positive, with 14.1% suspected cases. Suspected cases included children who were not currently testing positive; however, their parents had seen signs and symptoms of COVID-19. In addition, these children had at least one family member diagnosed with COVID-19.

As far as disease severity is concerned, 4.8% of cases were asymptomatic, 21.5% had mild disease, 26.9% had moderate disease, 38.2% had severe disease and 8.6% were critical cases. Of the children, 37.4% were admitted to the ICU, with 2.8% requiring mechanical ventilation. Fever was one of the common symptoms among the study population, present in 71.8% of children, followed by a cough (66.1%) and tachypnea (62.1%). X-ray abnormalities were reported in 81.1% of the children and 65.1% had elevated WBCs (Table 1). More than half (52.4%) of the study population required hospitalization between 8 and 14 days.

Figure 2 shows all the medication prescribed to the study population. Besides antibiotics, the three most commonly prescribed treatments were intravenous fluids and/or inhalation fluids (85.5%), antipyretics (61.7%) and corticosteroids (55.9%). 

A total of 370 (85.5%) neonates and children were prescribed antibiotics during their hospital stay, with 736 antibiotics prescribed among them (average 1.70 ± 0.98 antibiotics per patient; Table 2). More than half (54.3%) of the children were prescribed two antibiotics, followed by 24.9% with one, and 20.8% of the children were prescribed three antibiotics (Table 2). The majority of the antibiotics (75.5%) were administered parenterally. More than half of the antibiotics were prescribed for ≤5 days, and 39.4% for 6–10 days. 

Antibiotics prescribed according to the ATC class and subclass are shown in the Table 3. Third-generation cephalosporins and macrolides were the most commonly prescribed classes of antibiotics. The top three most commonly prescribed antibiotics were azithromycin (*n* = 183), ceftriaxone (*n* = 173) and meropenem (*n* = 74).

Antibiotic usage grouped according to the AWaRe classification are shown in Figure 3. Most of the prescribed antibiotics were from the ‘Watch’ group (80.4%, n = 592). Only 16.7% (n = 123) of the prescribed antibiotics were from the ‘Access’ group.

A comparison of antibiotic usage between the demographic and clinical variables are shown in Table 4. A significant difference regarding the extent and nature of antibiotics prescribed was observed among the different hospitals (*p* = 0.001). 

In a post hoc analysis (Table 5), the prescribing of antibiotics was significantly higher among neonates and children in Hospital 2 compared to those in Hospitals 3 and 4. As anticipated, antibiotic prescribing was substantially higher in COVID-19-positive patients, with an increasing number of antibiotics prescribed with increasing severity of COVID-19 (Table 4). Increased antibiotic prescribing was also observed among patients requiring mechanical ventilation, as well as those with elevated WBCs, CRP, D-dimer and ferritin levels (Table 4). 

A multiple linear regression analysis was performed to confirm all the factors linked with antibiotic prescribing among admitted children (Table 6). Increased COVID-19 severity, hospital length of stay and the hospital setting were found to be significantly associated with antibiotic prescribing.

As far as outcomes are concerned, 3.2% children died during the study period, whereas 96.8% were discharged from the participating hospitals.

Bacterial culture testing was ordered for 28 patients. Of these, 12 came back negative and 16 patients had identified bacterial pathogens. Consequently, the confirmed prevalence of bacterial or secondary bacterial infections was 3.7% (Figure 4). 

Details of culture and sensitivity testing are provided in Table 7. The most common bacterial pathogens identified in our study sample were *Pseudomonas aeruginosa* (*n* = 9, 32.1%) and *Streptococcus pneumoniae* (*n* = 3, 10.7%). 

## 3. Discussion

We believe this is one of the first studies conducted among neonates and children hospitalized in Pakistan due to suspected or confirmed COVID-19. This includes the clinical manifestations and laboratory findings of admitted children, the prevalence of bacterial co-infections and the subsequent management of children, including any prescribing of antimicrobials during their stay in hospitals. There was a lower prevalence of COVID-19 among admitted neonates and infants in this study compared with children in the higher age groups. This is similar to the findings in Ghana and India, as well as in the USA [9,12,61]. However, this is different to the findings from a study in Bangladesh, where the majority of admitted children were aged 0 to 5 years [10]. The low prevalence of COVID-19 in neonates and infants might be associated with low transmission levels in this age group, due to greater immobility. However, we cannot say this with certainty. 

The majority of admitted neonates and children in this study were COVID-19-positive and without co-morbidities, similar to findings from Ghana, Italy and India [9,12,62]. In addition, children were typically admitted with moderate-to-severe disease rather than asymptomatic or mild disease. This is similar to the findings of studies in Ghana, Iran and the USA [12,63,64]; however, different from a study in Saudi Arabia with very young infants (≤90 days), and Qatar, where thresholds for admitting children were lower than for adults, as well as in a study from Turkey [65,66,67]. The higher prevalence of severe and critical COVID-19 children reported in our study may be associated with the fact that we conducted this study among four referral/teachings hospitals, with mild cases likely to be treated in primary or secondary care hospitals in the province. 

More than one third of our study population was admitted to the ICU, with the most common symptoms among them being fever, cough and tachypnoea. This is similar to the US (25.5% admitted to ICU), as well as a systematic review comparing LMICs to higher-income countries [68,69]. However, our findings are different from studies in Bangladesh (13.7%), India (22.2%), Spain (19.7%) and the UK (4.1%) in terms of children admitted to ICU with COVID-19 [9,10,70,71]. The potential reasons for the high ICU admission in our study could again be referrals to these specialist hospitals. The symptoms of fever, cough and tachypnoea, typical of viral pneumonia, were similar through to other studies undertaken with children across countries [9,10,72,73,74]. X-ray abnormalities were one of the most frequent laboratory findings, followed by out-of-range WBCs and CRP in our study, which is similar to other published studies documenting changes in chest X-rays in patients with COVID-19 [3,74,75,76]. However, other authors found different findings [67]. We are not sure of the reasons behind the different findings between the studies. This may be due to differences in admittance criteria; however, we are not sure. 

There was appreciable prescribing of antibiotics among the neonates and children hospitalized with COVID19 in our study at >85%, with more than half being prescribed two antibiotics. In addition, nearly one-quarter of admitted patients were prescribed three different antibiotics during their hospital stay, with an average of 1.70 ± 0.98 antibiotics per patient. This is despite only 3.7% of neonates and children having confirmed bacterial co-infections or secondary bacterial infections. However, similarly high rates of antibiotic prescribing were seen in our previous study among all hospitalized neonates and children conducted between 10 December 2021 and 5 January 2022 [13], similar to studies undertaken in Bangladesh (86.3%) and India (75.3%) [9,10]. The prescribing of antibiotics in our study was though higher than seen in studies in Latin America (24.5%) and Turkey [67,77]. In addition, more than half of the antibiotics prescribed in our study were for ≤5 days and via parenteral route of administration. This is similar to a study from Spain, where the majority of the antibiotics administered were prescribed for <5 days among children hospitalized with COVID-19 [71]. 

Of equal concern is that 80.4% of the prescribed antibiotics were from the WHO ‘Watch’ category, increasing resistance rates [41]. This is similar though to the high rates of prescribing of ‘Watch’ antibiotics among children in Bangladesh and India [9,10]. These high rates need to be urgently addressed, given the already rising rates of AMR in Pakistan, and ongoing challenges to reduce this as part of their national action plan to reduce AMR [78,79]. We are seeing antimicrobial stewardship programs (ASPs) being successfully introduced among hospitals across LMICs to improve appropriate antibiotic prescribing despite initial concerns, and these can serve as exemplars for Pakistan in the future [44,45,80,81,82,83]. We will be following this up in future studies now that the AWaRe book has been published, providing guidance for a range of infections, combined with suggested prescribing and quality indicators [60,84].

The mortality rate of 3.2% seen in our study is lower than earlier studies in Pakistan involving children, which is encouraging [3,4]. This may reflect improved knowledge with treating children with COVID-19, with similar findings in other countries [10,85]. 

We are aware of a number of limitations with our study. Firstly, we only collected data from referral/tertiary care hospitals in the Punjab Province. Secondly, we did not use a validated data collection form. However, this was based on an appreciable number of published papers, as well as the experience of the co-authors in similar studies across countries, building on an earlier study involving neonates and children in Pakistan. Thirdly, we did not include any data on the use of anti-fungal medicines in our study despite ICU patients potentially having both fungal and bacterial infections. This is because we wanted to principally concentrate on antibiotic prescribing given increasing AMR rates in Pakistan, and the consequences. Lastly, we did not contact patients directly, and only collected data from the patients’ medical records. However, this is similar to all retrospective studies in hospitals, including point-prevalence surveys. Despite these limitations, we believe our findings are robust, providing direction to key stakeholder groups in Pakistan involved in managing neonates and children with COVID-19, especially surrounding the inappropriate prescribing of antibiotics. 

## 4. Materials and Methods

### 4.1. General Outline of the Study

A retrospective, medical records review study was conducted in the COVID-19 wards among four tertiary care hospitals in the Punjab Province, designated for the management of children suspected or diagnosed with COVID-19. Since the emergence of COVID-19 and identification of very first case of laboratory confirmed COVID-19 in the country, these hospitals have served as the referral hospitals for the province for the management of children with COVID-19, up to the age of 12 years. 

Punjab was selected for this study because it is the most populus province in Pakistan, with a number of previous studies undertaken by the co-authors in the province, including patients with COVID-19, serving as comparators [23,39,86,87,88,89,90]. Moreover, these tertiary care/teaching hospitals were equipped with adequate medical personnel, laboratory facilities and personal protective equipment, as well as medicines for the effective management of children admitted with COVID-19. Tertiary care hospitals were selected for the current study because they are likely to provide guidance to the primary and secondary care hospitals in this province and across Pakistan.

### 4.2. Study Variables

Based on previous studies [10,12,23,37,38,39,89,90,91,92], data on the following variables were collected from the medical records of hospitalized neonates and children suspected or confirmed with COVID-19:Demographic characteristics of the study participants, including their age, gender, residence, number of days at the hospital, admission to the intensive care unit, presence of any co-morbidity, and the use of ventilation during hospital stays.Status of COVID-19 was documented by the investigators. A positive or confirmed COVID-19 patient is defined as a positive result on a real-time reverse-transcriptase polymerase chain reaction (PCR) assay of nasal or pharyngeal swab specimens. As mentioned, suspected cases included children who were not currently testing positive; however, their parents had seen signs and symptoms of COVID-19. In addition, these children had at least one family member diagnosed with COVID-19.COVID-19 disease severity, categorized into asymptomatic, mild, moderate, severe and critical, as per the guidelines for the management of COVID-19 issued by the Ministry of National Health Services, Regulation, and Coordination, Government of Pakistan. Asymptomatic cases meant children were tested as positive with COVID-19; however, were currently without symptoms. Mild cases include those manifesting symptoms due to COVID-19, but without hemodynamic disturbances and X-ray abnormalities. Mild cases did not require oxygen and the oxygen saturation in mild cases must be ≥94%. Those patients that had abnormal chest X-rays, including X-rays infiltrates involving <50% of the total lung fields, oxygen saturation below 94%, but above 90%, and without any severe symptoms, were declared as moderate cases of COVID-19. Severe cases of COVID-19 included children that had a fever and cough along with respiratory rate < 30, severe respiratory distress, chest X-rays with infiltrates involving <50% of the total lung fields and oxygen saturation ≤ 90 on room air. Critical cases were those that showed a worsening of respiratory symptoms or the presence of acute respiratory distress syndrome (ARSD), respiratory or cardiac failure and bilateral opacities or lung collapse on chest X-rays or CT scans.Signs and symptoms of COVID-19 including a fever, cough, sore throat, tachypnea, deceases level of consciousness, headache/body ache, lethargy, nausea, vomiting, diarrhea, and irritability, were documented from individual medical records.Laboratory findings, including white blood cell counts (WBCs), c-reactive protein (CRP) level, D-dimer, and serum ferritin, were recorded by the investigators in patients’ notes. X-ray findings were reviewed by the medical doctors and consulted with treating physicians in case any clarification was needed. Normal ranges of WBCs, CRP D-dimer and serum ferritin were taken from the reference mentioned in the testing kits that were available and used in the laboratories of the participating hospitals.Existence of bacterial co-infection and secondary bacterial infection among COVID-19 patients. Bacterial co-infection was identified as those bacterial infections identified in ≤2 days after hospital admission due to COVID-19 and bacterial secondary infection as bacterial infections identified in >2 days after admission, confirmed microbiologically.Details about the antibiotics prescribed among hospitalized neonates and children. This included how many hospitalized COVID-19 patients were prescribed antibiotics during their stay in hospitals, as well as the existence of bacterial co-infection and bacterial secondary infections. Antibiotics were further classified according to the ATC classification, as well as the WHO AWaRe classification [60,93,94]. ‘Access’ antibiotics should typically be prescribed to treat commonly encountered infections, as they have a lower resistance potential, with those in the ‘Watch’ group ideally only prescribed in critical conditions, as they have a greater chance of resistance development. Those in the ‘Reserve’ category should only be prescribed in multi-drug resistance cases, with the aim of curbing rising AMR rates [41,60,84,94,95,96].The total number of antibiotics, the average number of antibiotics prescribed per patient, the duration of antibiotic therapy and the consumption of other antimicrobials. We did not specifically document the extent of prescribing of antifungals, although we are aware that there can be joint fungal and bacterial co-infections in patients in the ICU [22], as our main focus was on the extent of the prescribing of antibiotics, especially ‘Watch’ antibiotics in this population, alongside the extent of bacterial infections, including secondary bacterial infections.Outcomes, including whether neonates or children were discharged from a hospital or died.

### 4.3. Inclusion and Exclusion Criteria and Data Collection Procedures

Inclusion criteria included neonates and children ≤ 12 years age admitted to the COVID-19 wards, including intensive care units, among the four referral hospitals in Punjab, with suspected or proven COVID-19 and having complete medical records available at the hospital record room. 

All neonates and children admitted in other wards of the designated hospitals diagnosed with illnesses other than COVID-19 or having incomplete medical records were excluded from this study. 

The team of investigators comprised pharmacists, medical doctors, nurses, pharmacy and laboratory technicians. The investigators visited the record rooms for the COVID-19 wards in the participating hospitals and thoroughly searched the files to collect the necessary information. All the medical records were paper-based in these hospitals. 

### 4.4. Statictical Analyses

Continuous data were reported as the mean ± standard deviation, whereas categorical data were reported as the number (*N*) and/or percentage (%). Antibiotic use was compared among demographic and clinical variables using an independent t-test and ANOVA, where applicable. Furthermore, post hoc analyses were performed to determine potential differences between three or more groups when the ANOVA test yielded significant results. Multiple logistic regression analyses were also performed to determine factors linked with antibiotic use in children hospitalized with COVID-19 (method: stepwise). All data analysis were performed in SPSS version 22 for Microsoft windows.

### 4.5. Ethical Considerations

Ethical approval for the study was obtained from the Human Research Ethics Committee, Department of Pharmacy Practice, The University of Lahore (REC/DPP/FOP/49). Moreover, permission to conduct the study in the different hospitals was obtained from the administrators/ethic committee of each hospital prior to data collection. No personal patient information was collected at any stage of data collection. 

The confidentiality of the data was ensured by the allocation of anonymized identifier to every patient, which was separated from the data, for the purpose of verifying the accuracy of the recorded data. Institutional ethics committee allowed the investigators to collect data without patient consent, as this was a retrospective study with no direct contact with the patients and/or their parents. 

## 5. Conclusions

Overall, neonates and children admitted to tertiary hospitals in Pakistan due to COVID-19 were excessively prescribed antibiotics, despite very low prevalence of bacterial co-infections. Of equal concern was that the vast majority of prescribed antibiotics was from the ‘Watch’ category, potentially worsening AMR. This needs to be urgently addressed across Pakistan, with the appropriate implementation of ASPs. 

## Figures and Tables

**Figure 1 antibiotics-12-00646-f001:**
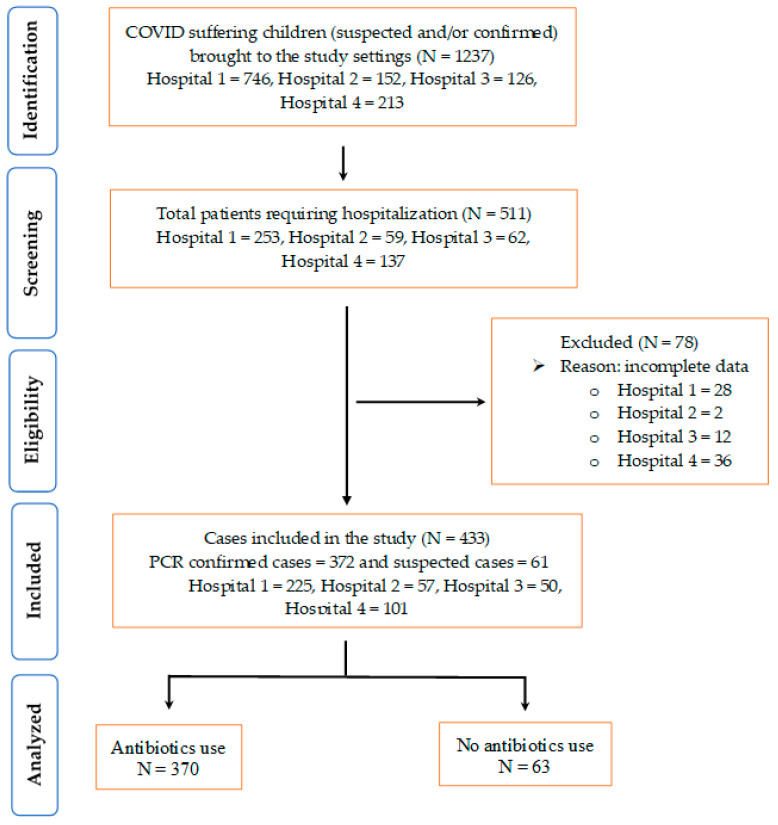
Flow chart of identification of COVID patients aged ≤ 12 for inclusion in the study.

**Figure 2 antibiotics-12-00646-f002:**
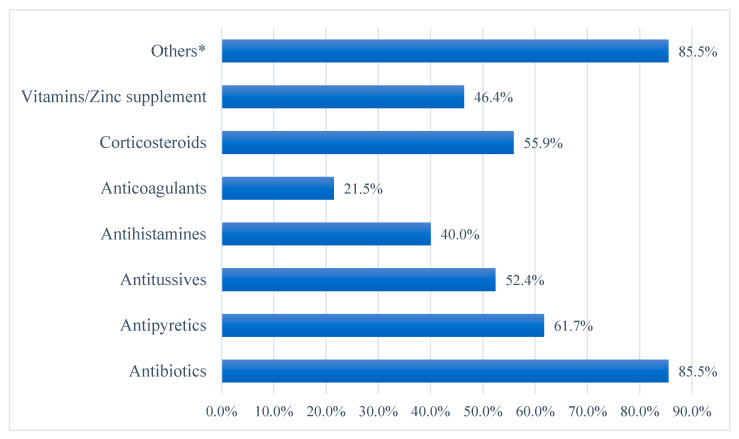
Details of medication prescribed to the study population. NB: * Intravenous fluids and/or inhalation fluids.

**Figure 3 antibiotics-12-00646-f003:**
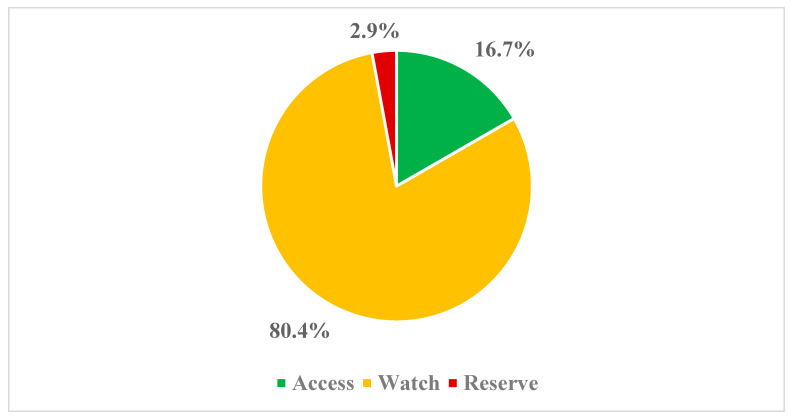
Antibiotic usage classified according to WHO AWaRe Classification. NB: Please refer to the methodology for the definitions of the AWaRe groupings.

**Figure 4 antibiotics-12-00646-f004:**
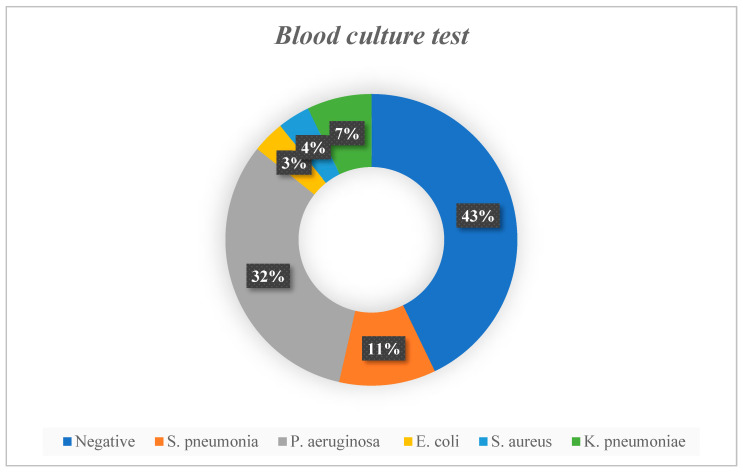
Findings of the blood culture tests (N = 28).

**Table 1 antibiotics-12-00646-t001:** Demographic and clinical characteristics of the sample.

Variables	*N* (%)
** *Number of patients per hospital* **	
H1	225 (52.0)
H2	57 (13.2)
H3	50 (11.5)
H4	101 (23.3)
** *Age* **	
Neonate (1–28 days)	17 (3.9)
Infant (1–12 months)	75 (17.3)
Toddler (1–5 years)	146 (33.7)
Child (5–12 years)	195 (45.0)
** *Gender* **	
Male	275 (63.5)
Female	158 (36.5)
** *Residence* **	
Rural	125 (28.9)
Urban	308 (71.1)
** *Comorbidity (including low birth weight, preterm and anemia)* **	
Yes	44 (10.2)
No	389 (89.8)
** *COVID status* **	
Positive	372 (85.9)
Suspected	61 (14.1)
** *Severity of COVID-19 (N = 372)* **	
Asymptomatic	18 (4.8)
Mild	80 (21.5)
Moderate	100 (26.9)
Severe	142 (38.2)
Critical	32 (8.6)
** *ICU admission* **	
Yes	162 (37.4)
No	271 (62.6)
** *Invasive mechanical ventilation* **	
Yes	12 (2.8)
No	421 (97.2)
** *Signs and symptoms* **	402 (92.8)
Tachypnea	269 (62.1)
Decreased level of consciousness	117 (27.0)
Cough	286 (66.1)
Fever	311 (71.8)
Sore throat	216 (30.5)
Headache/body aches	132 (30.5)
Lethargy	160 (37.0)
Nausea/vomiting	191 (44.1)
Diarrhea	76 (11.3)
Others	49 (11.3)
** *Lab and other findings* **	
Positive imaging findings	351 (81.1)
Elevated WBCs	282 (65.1)
Elevated CRP	164 (37.9)
Elevated D-dimer	93 (21.5)
Elevated ferritin	55 (12.7)
Bacterial culture testing	28 (6.5)
** *Length of stay* **	
≤7 days	112 (25.9)
8–14 days	227 (52.4)
15–21 days	74 (17.1)
>21 days	20 (4.6)

NB: ICU—Intensive Care Unit.

**Table 2 antibiotics-12-00646-t002:** Details of antibiotics used in the study population.

Variables	*N* (%)
** *Neonates and children prescribed antibiotics* **	
Yes	370 (85.5)
No	63 (14.5)
Total number of antibiotics prescribed to all the children prescribed antibiotics (*N* = 370)	736
Average antibiotics prescribed per patient (Mean ± SD)	1.70 ± 0.98
** *Number of antibiotics prescribed per patients (N = 370)* **	
One	92 (24.9)
Two	201 (54.3)
Three or more	77 (20.8)
** *Route of antibiotic therapy (N = 736)* **	
Intravenous	556 (75.5)
Oral	180 (24.5)
** *Duration of antibiotic therapy* **	
≤5 days	423 (57.5)
6–10 days	290 (39.4)
>10 days	23 (3.1)

**Table 3 antibiotics-12-00646-t003:** Details of prescribed antibiotics according to Anatomical Therapeutic Chemical (ATC) and AWaRe Classification.

ATC Class	Subclass	Name of the Antibiotic and ATC Class	AWaRe Class	N
Beta-lactam antibacterials, penicillins (J01C)	Penicillins with extended spectrum (J01CA)	Amoxicillin (J01CA04)	Access	2
		Ampicillin (J01CA01)	Access	11
		Co-amoxiclav (J01CR02)	Access	40
	Combinations of penicillins, including beta-lactamase inhibitors (J01CR)	Piperacillin + Tazobactam (J01CR05)	Watch	32
Other beta-lactam antibacterials (J01D)	Carbapenems (J01DH)	Meropenem (J01DH02)	Watch	74
	Third-generation cephalosporins (J01DD)	Ceftriaxone (J01DD04)	Watch	173
		Ceftazidime (J01DD02)	Watch	19
		Cephoperazone (J01DD12)	Watch	12
		Cefotaxime (J01DD01)	Watch	16
		Cefixime (J01DD08)	Watch	2
	Fourth-generation cephalosporins (J01DE)	Cefepime (J01DE01)	Watch	35
Macrolides, lincosamides and streptogramins (J01F)	Macrolides (J01FA)	Azithromycin (J01FA10)	Watch	183
		Clarithromycin (J01FA09)	Watch	5
Quinolone antibacterials (J01M)	Fluoroquinolones (J01MA)	Ciprofloxacin (J01MA02)	Watch	11
		Levofloxacin (J01MA12)	Watch	3
		Moxifloxacin (J01MA14)	Watch	2
Aminoglycoside antibacterials (J01G)	Other aminoglycosides (J01GB)	Amikacin (J01GB06)	Access	60
Other antibacterials (J01X)	Glycopeptide antibacterials (J01XA)	Vancomycin (J01XA01)	Watch	25
	Imidazole derivatives (J01XD)	Metronidazole (J01XD01)	Access	10
	Other antibacterials (J01XX)	Linezolid (J01XX08)	Reserve	21

NB: Details of the AWaRe classification and its importance are in methodology.

**Table 4 antibiotics-12-00646-t004:** Comparison of antibiotic usage among selected variables.

Variables	No. of Antibiotics	*p*-Value
**Hospitals**		
H1	1.81 ± 0.95	**0.001**
H2	2.11 ± 0.77	
H3	1.44 ± 1.03	
H4	1.35 ± 1.00	
**Age**		
Neonate (1–28 days)	1.53 ± 0.80	0.483
Infant (1–12 months)	1.69 ± 1.01	
Toddler (1–5 years)	1.79 ± 1.00	
Child (5–12 years)	1.65 ± 0.97	
**Gender**		
Male	1.70 ± 1.01	0.954
Female	1.70 ± 0.94	
**Residence**		
Rural	1.71 ± 0.97	0.869
Urban	1.69 ± 0.98	
**Comorbidity**		
Yes	1.84 ± 1.12	0.314
No	1.68 ± 0.96	
**COVID status**		
Positive	1.76 ± 0.98	**0.002**
Suspected	1.34 ± 0.89	
**Severity of COVID (*N* = 372)**		
Asymptomatic	0.44 ± 0.78	**<0.001**
Mild	1.05 ± 1.11	
Moderate	1.68 ± 0.68	
Severe	2.16 ± 0.65	
Critical	2.72 ± 0.81	
**ICU admission**		
Yes	2.22 ± 0.71	**<0.001**
No	1.39 ± 0.99	
**Invasive mechanical ventilation**		
Yes	2.67 ± 0.65	**<0.001**
No	1.67 ± 0.97	
**X-ray abnormalities**		
Yes	1.83 ± 1.88	**<0.001**
No	1.12 ± 1.17	
**Elevated WBCs**		
Yes	1.95 ± 0.86	**<0.001**
No	1.23 ± 1.02	
**Elevated CRP**		
Yes	2.19 ± 0.74	**<0.001**
No	1.41 ± 0.99	
**Elevated D-dimer**		
Yes	2.16 ± 0.74	**<0.001**
No	1.57 ± 1.00	
**Elevated ferritin**		
Yes	2.22 ± 0.81	**<0.001**
No	1.62 ± 0.98	
**Bacterial culture testing**		
Yes	1.96 ± 0.84	0.098
No	1.68 ± 0.99	
**Length of stay**		
≤7 days	1.04 ± 1.03	**<0.001**
8–14 days	1.73 ± 0.81	
15–21 days	2.28 ± 0.73	
>21 days	2.95 ± 0.39	

NB: Bold = statistically significant.

**Table 5 antibiotics-12-00646-t005:** Post hoc analysis of antibiotic use among hospital and COVID-19 severity categories.

Comparison	Mean Difference	Standard Error	*p*-Value
**Hospital**			
H1 vs. H2	−0.29	0.12	0.078
H1 vs. H3	0.37	0.16	0.098
H1 vs. H4	0.47	0.12	**0.001**
H2 vs. H3	0.67	0.18	**0.002**
H2 vs. H4	0.76	0.14	**<0.001**
H3 vs. H4	0.09	0.18	0.952
**COVID-19 severity**			
Asymptomatic vs. mild	−0.61	0.22	0.072
Asymptomatic vs. moderate	−1.24	0.20	**<0.001**
Asymptomatic vs. severe	−1.72	0.19	**<0.001**
Asymptomatic vs. critical	−2.27	0.23	**<0.001**
Mild vs. moderate	−0.63	0.14	**<0.001**
Mild vs. severe	−1.11	0.14	**<0.001**
Mild vs. critical	−1.67	0.19	**<0.001**
Moderate vs. severe	−0.48	0.09	**<0.001**
Moderate vs. critical	−1.04	0.16	**<0.001**
Severe vs. critical	−0.56	0.15	**0.007**
**Length of hospital stay (days)**			
≤7 vs. 8–14	−0.69	0.11	**<0.001**
≤7 vs. 15–21	−1.25	0.13	**<0.001**
≤7 vs. >21	−1.91	0.13	**<0.001**
8–14 vs. 15–21	−0.56	0.10	**<0.001**
8–14 vs. >21	−1.22	0.10	**<0.001**
15–21 vs. >21	−0.67	0.12	**<0.001**

NB: Bold = statistically significant.

**Table 6 antibiotics-12-00646-t006:** Factors associated with antibiotic use in children with COVID-19.

Coefficients ^a,b^							
Model	Unstandardized Coefficients		Standardized Coefficients	T	Sig.	95% CI for B	
	B	SE	Beta			Lower Bound	Upper Bound
(Constant)	0.010	0.149		0.068	0.946	−0.282	0.302
COVID-19 severity	0.404	0.044	0.427	9.226	<0.001	0.318	0.490
Length of hospital stay	0.370	0.058	0.293	6.325	<0.001	0.255	0.485
Hospital	−0.157	0.030	−0.204	−5.185	<0.001	−0.217	−0.098

^a^ Number of prescribed antibiotics. ^b^ Selecting only cases for which COVID-19 status = Positive.

**Table 7 antibiotics-12-00646-t007:** Findings of culture and sensitivity testing.

Pathogen	N	Sensitivity Results
*S. pneumonia*	3	Linezolid, levofloxacin, vancomycin
*P. aeruginosa*	4	Ceftriaxone, carbapenems
*P. aeruginosa*	2	Ceftriaxone, carbapenems, aminoglycosides
*P. aeruginosa*	3	Meropenem
*K. pneumoniae*	2	Pipericillin, ceftazidime, meropenem
*E. coli*	1	Pipericillin, metronidazole
*S. aureus*	1	Levofloxacin, imipenem

## Data Availability

Further data are available on reasonable request from the corresponding authors.

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
