# Peer review of "Antibiotic Overprescribing among Neonates and Children Hospitalized with COVID-19 in Pakistan and the Implications"

_antibiotics, 2023, doi:10.3390/antibiotics12040646_

Round 1

Reviewer 1 Report

Dear authors,

greetings of the day,

with interest i read your paper and found it well writen with good scientific soundness, the introduction result, and discussion were well described.

However, i don't know why you put results before the methods?

There are some old references :55,63,69,70,96.

The webography in the reference section must be completed by the date of assessment.

Best regards

Author Response

Comments and Suggestions for Authors

1) Dear authors, greetings of the day, with interest i read your paper and found it well written with good scientific soundness, the introduction result, and discussion were well described.

Author comments: Thank you for these good comments – much appreciated!

2) However, i don't know why you put results before the methods?

Author comments: Thank you for this – however, this is the style of the Journal. We hope this is OK with you!

3) There are some old references :55,63,69,70,96.

Author comments: Thank you – we have kept some of the references as important and changed others, e.g. ref 63 is one of the most robust references documenting the morbidity associated with AMR, Ref 70 documents some of the resistance patterns in Pakistan and 96 refers to the challenges of implementing ASPs in hospitals in LMICs – now though being overcome. We hope this is acceptable.

4) The webography in the reference section must be completed by the date of assessment

Author comments: Thank you – now addressed

Reviewer 2 Report

It would be helpful if the authors could provide evidence of Antimicrobial Resistance and how they concluded that what they observed was due to AMR.

Author Response

Comments and Suggestions for Authors

It would be helpful if the authors could provide evidence of Antimicrobial Resistance and how they concluded that what they observed was due to AMR.

Author comments: Thank you for this. We have extensively documented from a range of papers that the increased prescribing of antibiotics in patients with COVID-19 increases AMR along with the excessive prescribing of Watch and Reserve antibiotics over Access antibiotics. Unfortunately in Pakistan CST testing is rare compared with higher income countries – so we cannot show this specifically in the hospitals treating our patients. However – as mentioned – the inference is there from the robust quoted papers. We hope this is acceptable to you.

Reviewer 3 Report

Dear Authors

I would like to thank you for the opportunity of reviewing this interesting paper that is focused on a very remarkable and challenging topic that is a lively argument also in daily clinical practice. Although it has been more than 2 years since the first outbreak, the coronavirus disease 2019 (COVID-19) pandemic is still having a profound and devastating impact on global healthcare systems. COVID-19 is responsible for a respiratory disease whose broad spectrum of severity ranges from asymptomatic or mildly symptomatic infection to severe bilateral pneumonia, which may lead to acute respiratory distress syndrome (ARDS), requiring non-invasive or invasive mechanical ventilation and Intensive Care Unit (ICU) admission. Several complications can arise during ICU stay, from both COVID-19 extensive lung damage and extra-pulmonary involvement, as well as those secondary to mechanical supporting systems. Among these, bacterial co-infections and superinfections play an important role in COVID-19 disease and have been associated with increased morbidity and mortality, especially in critically ill patients.

The present article aimed to track bacterial coinfections and superinfections among neonates and children hospitalized in Pakistan due to COVID-19, investigating the main clinical manifestations and laboratory findings. Moreover, the present study aimed to assess the prevalence of antibiotic prescribing among this particular population in order to guide future therapeutic strategies. In particular, the Authors demonstrated that neonates and children admitted to tertiary hospitals in Pakistan due to COVID-19 were excessively treated with antibiotics despite the very low prevalence of bacterial co-infections. Moreover, the vast majority of prescribed antibiotics were in the “Watch category” list, thus they could potentially worsen antimicrobial resistance. 

This paper is pleasurable to read, although it suffers from some limitations that Authors can easily adjust in order to improve their review making it more eligible for this important Journal. Furthermore, the Authors can improve some sections of the paper, adding information and including other important references about this topic that, in my opinion, should be cited and discussed. 

Although the language used is quite appropriate, I (I am not a native English speaker) recommend to the Authors to obtain a certified native speaker with proficiencies in the scientific-medical field to complete properly this paper (if not yet done). Moreover, I recommend making a further revision of the manuscript to fix some small typing/language errors. For example, “Conclusion” needs to be checked and corrected “Overall, neonates and children admitted to tertiary hospitals in Pakistan due to COVID-19 are excessively MISSING VERB antibiotics despite THE very low prevalence of bacterial co-infections.”.

The title is clear and direct. However, from a stylistic point of view, I believe it could be improved and more focused on the results. For example, “Antibiotic overprescribing in neonates and children hospitalized with COVID-19 in Pakistan.”

The Authors did not correctly report keywords from MeSH Browser. In particular, I checked for example “AWaRe classification” on MeSH Browser and this is not a KW. This is important, in my personal opinion, in order to increase the traceability of this paper (and consequently the possibility of the Journal being cited by Readers and Stakeholders). I suggest the check of all KW. 

Although the introduction fits the context of the study, lines 43-67 are a bit redundant, and the text seems too long, please reduce it.

Moreover, many concepts could be more clearly explicated in an exhaustive introduction, which would help readers to become passionate about reading the paper and using it as a reference. After introducing the topic of children affected with COVID-19, it is important to underline that despite viral pneumonia has been recognized as the main clinical presentation of this disease, representing the main cause of its severity and mortality, COVID-19 infection can cause several complications also in other organs, with coagulation disorders (pulmonary embolism, venous thromboembolism, hemorrhages and acute ischemic stroke) and abdominal involvement (acute mesenteric ischemia, pancreatitis and acute kidney injury), especially in severely ill patients and those admitted to ICU, even in children [Diagnostics (Basel). 2022;12(4):846. doi:10.3390/diagnostics12040846]. Then, the topic regarding the co- and super-infections in COVID-19 patients can be introduced, specifying how ICU patients with COVID-19 are susceptible to other bacterial, viral and fungal infections [Diagnostics (Basel). 2022;12(7):1617. doi: 10.3390/diagnostics12071617] [Eur J Clin Microbiol Infect Dis. 2021;40(3):495-502. doi: 10.1007/s10096-020-04142-w]. Please, cite these articles and introduce these important aspects in this section. Only after making these important clarifications, the Authors should finally address the issue of antibiotic overprescribing in this population. Lines 85-106 should be reduced as they are redundant. 

Finally, please be clearer regarding the aims of the study. For example “The first aim of the present study was to investigate the clinical manifestations and laboratory findings among neonates and children hospitalized in Pakistan due to COVID-19. Secondly, the prevalence of antibiotic prescribing among those children with a concomitant bacterial co-infection or superinfection was assessed.”

Figure 1 should be reformatted in order to facilitate the understanding of which patients were excluded and which ones were included.

In Table 1, please specify which “Comorbidity” was considered; in addition, “X-Ray abnormalities” should be corrected with “positive imaging findings”.

In “Material and Methods”: please remove lines 289-296. Moreover, “Based on previous studies [15,20,49-52,101-104]” is not necessary, since the text well-explain which data and how they were collected, thus remove it. I don’t believe that “suspected cases” should be included in the analysis and these patients should be removed to avoid false results.

When you describe imaging findings, please use these references [Radiol Cardiothorac Imaging. 2020 Jun 11;2(3):e200213. doi: 10.1148/ryct.2020200213] [Cureus. 2022;14(1):e21656. doi:10.7759/cureus.21656].

The main limitation of the study is that no other concomitant viral and/or fungal infections other than those caused by bacteria were reported. This data is essential given that the probability of survival could be influenced by the presence of any other microorganism. Indeed, it is known that ICU patients with COVID-19 often present with a concomitant fungal AND bacterial infection. [Chest. 2021 Aug;160(2):454-465. doi: 10.1016/j.chest.2021.04.002] [J Med Virol. 2022 May;94(5):1920-1925. doi: 10.1002/jmv.27548]. If possible, therefore, the Authors should include these data in the analysis. If this data is not available, it is necessary to include this important limit at the end of the discussion.

Finally, I think references should be reformatted as suggested by Antibiotics Author’s guidelines (Author 1, A.B.; Author 2, C.D. Title of the article. Abbreviated Journal Name YearVolume, page range)

Best regards, 

Author Response

Comments and Suggestions for Authors

1) I would like to thank you for the opportunity of reviewing this interesting paper that is focused on a very remarkable and challenging topic that is a lively argument also in daily clinical practice. Although it has been more than 2 years since the first outbreak, the coronavirus disease 2019 (COVID-19) pandemic is still having a profound and devastating impact on global healthcare systems. COVID-19 is responsible for a respiratory disease whose broad spectrum of severity ranges from asymptomatic or mildly symptomatic infection to severe bilateral pneumonia, which may lead to acute respiratory distress syndrome (ARDS), requiring non-invasive or invasive mechanical ventilation and Intensive Care Unit (ICU) admission. Several complications can arise during ICU stay, from both COVID-19 extensive lung damage and extra-pulmonary involvement, as well as those secondary to mechanical supporting systems. Among these, bacterial co-infections and superinfections play an important role in COVID-19 disease and have been associated with increased morbidity and mortality, especially in critically ill patients.

Author comments: Thank you for these kind comments – very much appreciated!

2) The present article aimed to track bacterial coinfections and superinfections among neonates and children hospitalized in Pakistan due to COVID-19, investigating the main clinical manifestations and laboratory findings. Moreover, the present study aimed to assess the prevalence of antibiotic prescribing among this particular population in order to guide future therapeutic strategies. In particular, the Authors demonstrated that neonates and children admitted to tertiary hospitals in Pakistan due to COVID-19 were excessively treated with antibiotics despite the very low prevalence of bacterial co-infections. Moreover, the vast majority of prescribed antibiotics were in the “Watch category” list, thus they could potentially worsen antimicrobial resistance. 

Author comments: Thank you for this summary – appreciated.

3) This paper is pleasurable to read, although it suffers from some limitations that Authors can easily adjust in order to improve their review making it more eligible for this important Journal. Furthermore, the Authors can improve some sections of the paper, adding information and including other important references about this topic that, in my opinion, should be cited and discussed. 

Author comments: Thank you for your help – we hope we have now adequately improved the paper and it is now OK.

4) Although the language used is quite appropriate, I (I am not a native English speaker) recommend to the Authors to obtain a certified native speaker with proficiencies in the scientific-medical field to complete properly this paper (if not yet done). Moreover, I recommend making a further revision of the manuscript to fix some small typing/language errors. For example, “Conclusion” needs to be checked and corrected “Overall, neonates and children admitted to tertiary hospitals in Pakistan due to COVID-19 are excessively MISSING VERB antibiotics despite THE very low prevalence of bacterial co-infections.”.

Author comments: Thank you for this. We have now updated the paper with the help of one of the co-authors who is a native English speaker with over 500 publications in peer-reviewed Journals since 2008. We hope this is now acceptable

5) The title is clear and direct. However, from a stylistic point of view, I believe it could be improved and more focused on the results. For example, “Antibiotic overprescribing in neonates and children hospitalized with COVID-19 in Pakistan.”

Author comments: Thank you for this – now updated. We hope this is now OK.

6) The Authors did not correctly report keywords from MeSH Browser. In particular, I checked for example “AWaRe classification” on MeSH Browser and this is not a KW. This is important, in my personal opinion, in order to increase the traceability of this paper (and consequently the possibility of the Journal being cited by Readers and Stakeholders). I suggest the check of all KW. 

Author comments: Thank you for the comment and pointing out it. We have updated keywords as per MeSH Browser as well as kept the AWaRe classification in view of its growing importance directing antibiotic utilization in the future. We hope it will be acceptable now.

7) Although the introduction fits the context of the study, lines 43-67 are a bit redundant, and the text seems too long, please reduce it. Moreover, many concepts could be more clearly explicated in an exhaustive introduction, which would help readers to become passionate about reading the paper and using it as a reference. After introducing the topic of children affected with COVID-19, it is important to underline that despite viral pneumonia has been recognized as the main clinical presentation of this disease, representing the main cause of its severity and mortality, COVID-19 infection can cause several complications also in other organs, with coagulation disorders (pulmonary embolism, venous thromboembolism, hemorrhages and acute ischemic stroke) and abdominal involvement (acute mesenteric ischemia, pancreatitis and acute kidney injury), especially in severely ill patients and those admitted to ICU, even in children [Diagnostics (Basel). 2022;12(4):846. doi:10.3390/diagnostics12040846]. Then, the topic regarding the co- and super-infections in COVID-19 patients can be introduced, specifying how ICU patients with COVID-19 are susceptible to other bacterial, viral and fungal infections [Diagnostics (Basel). 2022;12(7):1617. doi: 10.3390/diagnostics12071617] [Eur J Clin Microbiol Infect Dis. 2021;40(3):495-502. doi: 10.1007/s10096-020-04142-w]. Please, cite these articles and introduce these important aspects in this section. Only after making these important clarifications, the Authors should finally address the issue of antibiotic overprescribing in this population. Lines 85-106 should be reduced as they are redundant.

Author comments: Thank you for this. We have now appreciably revised the Introduction as directed and included these references and others regarding Pediatric Inflammatory Multisystem Syndrome. We hope this is now acceptable.

8) Finally, please be clearer regarding the aims of the study. For example, “The first aim of the present study was to investigate the clinical manifestations and laboratory findings among neonates and children hospitalized in Pakistan due to COVID-19. Secondly, the prevalence of antibiotic prescribing among those children with a concomitant bacterial co-infection or superinfection was assessed.”

Author comments: Thank you. We have updated the aims as directed, and hope this is now acceptable.

9) Figure 1 should be reformatted in order to facilitate the understanding of which patients were excluded and which ones were included.

Author comments: Thank you. We have now updated Figure 1. Hopefully, it will be acceptable now.

10) In Table 1, please specify which “Comorbidity” was considered; in addition, “X-Ray abnormalities” should be corrected with “positive imaging findings”.

Author comments: Thank you. We have updated this in the revised version of manuscript.

11) In “Material and Methods”: please remove lines 289-296. Moreover, “Based on previous studies [15,20,49-52,101-104]” is not necessary, since the text well-explain which data and how they were collected, thus remove it. I don’t believe that “suspected cases” should be included in the analysis and these patients should be removed to avoid false results.

Author comments: Thank you for this. However, Antibiotics, other similar Journals and Journals, like to see justification for the methods – hence we would like to keep these references if we can. We hope this is OK with you.

Suspected cases were those that were not positive but the patients had sign and symptoms of COVID-19 positive. In addition, these patients had one family member diagnosed COVID-19. Moreover, the inefficiency of laboratory to detect the SARS-Cov was another factor. These cases were treated as COVID-19 positive and prescribed antibiotics. We have revised the text accordingly, and hope this is now OK.

12) When you describe imaging findings, please use these references [Radiol Cardiothorac Imaging. 2020 Jun 11;2(3):e200213. doi: 10.1148/ryct.2020200213] [Cureus. 2022;14(1):e21656. doi:10.7759/cureus.21656].

Author comments: Thank you – now included.

13) The main limitation of the study is that no other concomitant viral and/or fungal infections other than those caused by bacteria were reported. This data is essential given that the probability of survival could be influenced by the presence of any other microorganism. Indeed, it is known that ICU patients with COVID-19 often present with a concomitant fungal AND bacterial infection. [Chest. 2021 Aug;160(2):454-465. doi: 10.1016/j.chest.2021.04.002] [J Med Virol. 2022 May;94(5):1920-1925. doi: 10.1002/jmv.27548]. If possible, therefore, the Authors should include these data in the analysis. If this data is not available, it is necessary to include this important limit at the end of the discussion.

Author comments: Thank you for the suggestion. We did not include antifungal agents for the reasons stated although now included this reference and updated the limitations section. We will work on this in the future, and hope this is now acceptable.

14) Finally, I think references should be reformatted as suggested by Antibiotics Author’s guidelines (Author 1, A.B.; Author 2, C.D. Title of the article. Abbreviated Journal Name YearVolume, page range)

Author comments: Thank you – we will work with the Journal to update the references as undertaken in previous papers that we have published with Antibiotics. We hope this is OK with you.

Round 2

Reviewer 3 Report

Authors addressed raised points appropriately.